# *Symbiodiniaceae* and *Ruegeria* sp. Co-Cultivation to Enhance Nutrient Exchanges in Coral Holobiont

**DOI:** 10.3390/microorganisms12061217

**Published:** 2024-06-17

**Authors:** Yawen Liu, Huan Wu, Yang Shu, Yanying Hua, Pengcheng Fu

**Affiliations:** State Key Laboratory of Marine Resource Utilization in South China Sea, Hainan University, Haikou 570228, China; 21110832000005@hainanu.edu.cn (Y.L.); 21210707030016@hainanu.edu.cn (H.W.); 20083200110014@hainanu.edu.cn (Y.S.); 22220951350025@hainanu.edu.cn (Y.H.)

**Keywords:** coral, isotope labeling, probiotics, Raman spectra, single cell, *Symbiodiniaceae*

## Abstract

The symbiotic relationship between corals and their associated microorganisms is crucial for the health of coral reef eco-environmental systems. Recently, there has been a growing interest in unraveling how the manipulation of symbiont nutrient cycling affects the stress tolerance in the holobiont of coral reefs. However, most studies have primarily focused on coral–*Symbiodiniaceae*–bacterial interactions as a whole, neglecting the interactions between *Symbiodiniaceae* and bacteria, which remain largely unexplored. In this study, we proposed a hypothesis that there exists an inner symbiotic loop of *Symbiodiniaceae* and bacteria within the coral symbiotic loop. We conducted experiments to demonstrate how metabolic exchanges between *Symbiodiniaceae* and bacteria facilitate the nutritional supply necessary for cellular growth. It was seen that the beneficial bacterium, *Ruegeria* sp., supplied a nitrogen source to the *Symbiodiniaceae* strain *Durusdinium* sp., allowing this dinoflagellate to thrive in a nitrogen-free medium. The *Ruegeria* sp.–*Durusdinium* sp. interaction was confirmed through ^15^N-stable isotope probing–single cell Raman spectroscopy, in which ^15^N infiltrated into the bacterial cells for intracellular metabolism, and eventually the labeled nitrogen source was traced within the macromolecules of *Symbiodiniaceae* cells. The investigation into *Symbiodiniaceae* loop interactions validates our hypothesis and contributes to a comprehensive understanding of the intricate coral holobiont. These findings have the potential to enhance the health of coral reefs in the face of global climate change.

## 1. Introduction 

Algae, commonly recognized as major components of phytoplankton, play a crucial role as primary producers in aquatic ecosystems. They contribute to approximately 50% of the biospheric net primary productivity and represent approximately 0.2% of the global primary producer biomass [1]. The dissolved organic carbon from phytoplankton exudation serves as a major energy source to drive heterotrophic prokaryote respiration and growth. In return, inorganic nutrients produced by heterotrophic prokaryotes enhance mineralization for phytoplankton [2,3,4].

Within marine ecosystems, coral hosts form an endosymbiosis with their microbiota, comprising *Symbiodiniaceae*, fungi, bacteria, archaea and viruses, and more, to create the coral holobiont. Among them, *Symbiodiniaceae* are photosynthetic dinoflagellates, which are also colloquially known as *zooxanthellae*. They are single-cell microalgae and constitute the phototrophic components of the coral holobiont [5,6,7]. The photosynthetic products of *Symbiodiniaceae* are transferred to the coral host, supplying them as a source of energy, which is vital for coral growth and the construction of calcium carbonate skeletons. Corals assist *Symbiodiniaceae* by providing carbon dioxide and trace elements through the marine environment, and heterotrophically fix the photosynthetically derived organic carbon by *Symbiodiniaceae* [8]. Simultaneously, symbiotic bacteria play a pivotal role in maintaining the health of the coral holobiont and facilitating the adaptations of this intricate biological system in the ocean [9]. Numerous studies have demonstrated that symbiotic microalgae and bacteria act in concert to maintain coral in homeostasis, as any destruction in either component results in the dissociation of the coral holobiont [10,11,12].

The mechanism of symbiotic relationships between microalgae and bacteria in the coral holobiont was confirmed by earlier studies, which elucidated that algal photosynthesis provides a high concentration of molecular oxygen (>200% saturation), aiding coral and associated prokaryotic microorganisms in respiration and biosynthesis [13,14]. The bacterial community residing in coral skeletons has been estimated to fulfill 50% of the total nitrogen requirements of their symbiotic partners, with organic compounds produced by cyanobacteria aiding the coral tissue [15]. On the other hand, bacteria produce antibiotics to protect their hosts and *Symbiodiniaceae* from the diseases caused by pathogens, thereby maintaining symbiotic health and improving its resilience to environmental stresses, ultimately preventing coral bleaching [16,17]. Bacterial communities in coral holobionts have been proven to be beneficial for the animal hosts and *Symbiodiniaceae*, in particular, as the diazotrophs to provide nitrogen sources. Therefore, the investigation of *Symbiodiniaceae*–bacteria interactions becomes a logical step toward understanding the mechanisms underlying the intricate multi-partner associations that occur within the coral holobiont.

Recent advancements in deep-sequencing methodologies have been employed to explore the impact of bacterial metabolism on *Symbiodiniaceae*’s nutrition and survival [18]. Over the decades, the development of rapid and cost-effective sequencing technologies has provided microbiologists with access to genome fragments and even the complete genomes of marine microbes. Despite the availability of such rich genomic information, the interpretation of the metabolite exchange within coral holobionts from the omic data remains difficult.

Currently, secondary ion mass spectrometry (NanoSIMS) has been applied to reveal the transfer of carbon (C) and nitrogen (N) among coral, *Symbiodiniaceae* and microorganisms by measuring 13C and 15N enrichment at the single-cell level [19]. However, mass spectroscopy, including NanoSIMS, is a destructive technique, preventing cells of interest from further studies, such as single-cell genomics and even cultivation. Single-cell Raman spectroscopy (SCRS) refers to the collection of Raman spectra obtained from individual cells. Raman spectroscopy is a non-destructive technique that measures the vibrational modes of molecules in a sample, providing information about its molecular composition and the structure of a single cell. In microbiology, this technique has been reported, for example, for the rapid spectroscopic identification of bacteria and fungi [20], and for studying bacterial metabolism and interactions at single-cell level [21]. The SCRS technique uses laser light to generate a chemical fingerprint of a single cell, and can identify different metabolic phenotypes of cells based on the Raman peaks in the spectra. Analyzing the Raman spectra of multiple cells in a population or symbiosis allows for the creation of a ramanome, which represents a metabolic snapshot of the population. A spectrum analysis of microbial cells, although more intricate than biofilms, facilitates the identification of several compounds, such as phenylalanine [22], tryptophan [23], carotenoids [24] and cytochrome c [25].

Isotope-labeling techniques, combined with SCRS, provide a tool to measure the cellular intake rates of substrates [26,27], monitor the biosynthetic profiles of cells, such as carotenoids, proteins and triacylglycerols [28], and characterize cellular responses to environmental changes [29].

In this study, we proposed a hypothesis that there exists an inner symbiotic loop of *Symbiodiniaceae* and bacteria (*Symbiodiniaceae* loop) within the coral symbiotic loop (coral–*Symbiodiniaceae*–bacteria). While previous coral studies have predominantly focused on the holistic interactions involving corals, *Symbiodiniaceae*, and bacteria, the nuanced relationship between *Symbiodiniaceae* and bacteria has been largely unexplored. Notably, we delved into the lesser-explored *Symbiodiniaceae* loop interactions to demonstrate how metabolic exchanges in the *Symbiodiniaceae* loop facilitate the nutritional supply necessary for cellular growth. We conducted a comprehensive screening and an in vitro cultivation of symbiotic microorganisms isolated from the reef-building corals *Acropora hyacinthus* [30] and *Galaxea fascicularis* [31], which resulted in the identification of a *Ruegeria* sp. strain containing nitrogen fixation genes. We evaluated the photosynthetic efficiency of *Symbiodiniaceae* under varying concentrations of *Ruegeria* sp. MR31c using chlorophyll fluorescence in vivo, as well as the accumulation of main photosynthetic pigments. Our investigation further substantiated the role of this beneficial bacterium in supplying nitrogenous compounds to *Durusdinium* sp. for intracellular nutrition, as confirmed through ^15^N-stable isotope probing–single-cell Raman spectroscopy (^15^N-SIP-SCRS). This technique unveiled the infiltration of ^15^N into bacterial cells, where it was utilized for intracellular metabolism. Eventually, we traced the labeled nitrogen source within the macromolecules, including proteins, carbohydrates and lipids, within the *Symbiodiniaceae* cells.

## 2. Materials and Methods

### 2.1. Sampling Procedures

*A. hyacinthus* and *G. fascicularis* corals were collected at the Wuzhizhou island, Sanya, Hainan, China (18°18′52.8″ N 109°46′07.9″ E). These two coral species were transported in sterile plastic bags and then packed in Styrofoam boxes containing 1 L of seawater and sent by car to a laboratory. Upon arrival at the research station, the coral colonies were fragmented using a pair of pliers (Maxspect, Ltd., Hongkong, China) in 5 cm fragments.

### 2.2. Isolation of Bacterial Strains and Symbiodiniaceae from Corals

Three previously tagged fragments of *A. hyacinthus* and *G. fascicularis* were used as a material to isolate bacterial and *Symbiodiniaceae* strains. Two approaches were used for microbial isolation. First, sterile seawater was used to rinse the coral surface to remove impurities. After cleaning, coral fragments were placed in a sterile mortar and continuously ground with sterile seawater until they were sufficiently fine, then transferred to a 50 mL tube and subsequently centrifuged at 5000× *g* for 5 min. Triplicate samples (50–100 μL) were inoculated into Petri dishes containing 20 mL of Marine Agar 2216E medium (19.45 g NaCl, 8.8 g MgCl_2_, 5 g tryptone, 3.24 g Na_2_SO_3_, 1.8 g CaCl_2_, 1 g yeast extract, 0.55 g KCl, 0.16 g NaHCO_3_, 0.1 g FeCl_2_, 0.08 g KBr, 0.03 g SrCl_2_, 0.02 g H_3_BO_3_, and 15 g of agar in 1 L distilled water at pH 7.6 ± 0.2), 2.5% NaCl Luria-bertani medium (10 g tryptone, 5 g yeast extract, 25 g NaCl and 15 g of agar in 1 L distilled water), sea water medium (1000 mL of seawater and 15 g of agar), SaltCzapek-DoxAgar (CDA) medium (2 g NaNO_3_, 1 g K_2_HPO_4_, 0.5 g MgSO_4_, 0.5 g KCl, 0.01 g FeSO_4_, 30 g Sucrose, and 15 g of agar in 1 L distilled water) and L1 medium (75 μg L^−1^ NaNO_3_, 5 μg L^−1^ NaH_2_PO_4_·H_2_O, trace elements and Vitamin) with antibiotics [32] (antibiotic composition: final concentration of 50 mg L^−1^ ampicillin, kanamycin, chloramphenicol, chlortetracycline, streptomycin and penicillin). In addition, the centrifuged pellet was resuspended in sterile seawater and triplicate subsamples (50–100 μL) of 10^−1^, 10^−2^, 10^−3^, 10^−4^, 10^−5^ dilutions were inoculated by the above medium. All the plates were incubated at 26 °C for 72 h, and followed day/night cycles (12 h/12 h) with 60 μmol of photons m^−2^ s^−1^ from 07:00 to 19:00 daily. A total of 42 colonies were isolated, and, based on the colony color and morphology, 24 strains were derived from milled slurries and 15 were derived from resuspensions. Each morphologically different colony was snap frozen in liquid nitrogen and stored in −80 °C fridge with a final concentration of 25% glycerol, and recovered when necessary for screening.

### 2.3. Functional Screening for Probiotic Bacteria Using 16S/ITS2 rDNA Gene Sequencing

Depending on the required biomass, *Symbiodiniaceae* cultures were grown in 6-well plates for the suspended cell cultures (3516, Corning, Corning, NY, USA), sealed with parafilm to prevent evaporation, and in 250 mL polycarbonate flasks for the purified single colonies with vented caps (17211, Beijing labgic technology Co., Ltd., Beijing, China). Bacterial genomes were isolated using TIANamp Bacteria DNA Kit (TianGen Biotech Co., Ltd., Beijing, China), following the manufacturer’s instructions. The DNA concentration was determined using the micro ultraviolet spectrophotometer (Nanodrop-2000, Thermo Fisher, Waltham, MA, USA). The purified 16S rRNA gene was amplified using the primers 27F (5′-AGAGTTTGATCCTGGCTCAG-3′) and 1492R (5′-TACCTTGTTACGACTT-3′) by 2 × Taq PCR master mix (TianGen Biotech Co Ltd., Beijing, China) and polymerase chain reaction (PCR). PCR cycles included 3 min of pre-denaturation at 94 °C, 30 cycles of 94 °C for 40 s, 55 °C for 1 min, 72 °C for 2 min and a final extension cycle of 10 min at 72 °C. *Symbiodiniaceae* genomes were isolated using the Hi-DNA secure Plant Kit (TianGen Biotech Co Ltd., Beijing, China), following the manufacturer’s instructions. To amplify the fungi ITS2 rDNA gene, the primers forward (5′-ATCGATGAAGAACGCAGC-3′) and reverse (5′-TCCTCCGCTTATTGATATGCCCCG-3′) were used. The thermal cycler conditions were as follows: initial denaturation at 95 °C for 3 min, 30 cycles of 95 °C for 30 s, 53 °C for 30 s, and 72 °C for 2 min, followed by a final extension at 72 °C for 5 min. Next, 5 μL of each PCR product was run on a 1% agarose gel to confirm successful amplification. PCR products were sequenced on the 3730xl DNA analyzer (Illumina PE150, Guangzhou, China) platform. The sequences were quality trimmed using Sequencher 4.6 and analyzed by BLAST of National Biotechnology Information Center. They were initially aligned using ClustalW as implemented in MegaX.

Each phenotypically distinct strain of bacteria was screened for beneficial traits for corals, according to Peixoto et al. [33]. Nitrogen cycle genes, such as dinitrogenase reductase genes (nifH) and nitrite reductase genes (*nir*K and *nir*S), were screened from genomic DNA samples by PCR (primer and PCR cycling information are shown in the Appendix A). The PCR amplification products were subjected to agarose gel electrophoresis, and bacteria with nitrogen-fixing genes were screened out as potential probiotic bacteria of *Symbiodiniaceae*.

### 2.4. Symbiodiniaceae-Bacteria Co-Culture

Bacterial were inoculated into 50 mL flasks with L1 medium and cultivated at 25 °C for 48 h, with shaking at 150 rpm. The cells were centrifuged at 10,000 rpm, washed twice with sterile seawater, and transferred to fresh nitrogen-free L1 medium to reach a level of absorbance at 600 nm of 0.005, 0.01, 0.05, 0.1, 0.5. *Symbiodiniaceae* was inoculated into 200 mL of nitrogen-free L1 medium containing different concentrations of bacteria with a density of 105 cells mL^−1^. The algal growth in the culture was determined by the daily detection of cell density and chlorophyll fluorescence, and photosynthetic pigments were measured within 14 days. The number of algal cells was measured by direct counting under a microscope using a Sedgwick-Rafter counting chamber, and their abundance was calculated based on the sample volume. The photochemical efficiency of the AG11 was assessed using pulse amplitude-modulated (PAM) fluorometry (Dual-PAM-100, WALZ, Effeltrich, Germany). The control experiment included a bacterial culture with sterile nitrogen-free L1 medium and mixed *Symbiodiniaceae*–bacterial cultures, in which the initial *Symbiodiniaceae* concentration was 105 cells mL^−1^.

### 2.5. Labeling Bacteria with Stable Isotope ^15^N-NH_4_Cl

The nutrient exchanges between *Symbiodiniaceae* and bacteria were determined using ^15^N-labelled nitrogen sources. For this, 75 mg mL^−1^ of ^15^N-NH_4_Cl was added to L1 medium and cultured with a constant temperature shaker (THZ-D, Shenglan, Ltd., Changzhou, China) for 48 h, with shaking at 150 rpm. The control group was added with the same concentration of NH_4_Cl.

### 2.6. SIP-Bacteria and Symbiodiniaceae Co-Culture

Bacterial cultures were centrifuged at 10,000 rpm, washed twice with sterile seawater and transferred to fresh nitrogen-free L1 medium to reach a level of absorbance at 600 nm of 0.5. Bacteria and *Symbiodiniaceae* were co-cultured in the same manner as in 2.4. After 72 h, the signals of microalgae and bacteria were detected by single-cell Raman, respectively.

### 2.7. Determination of Photosynthetic Pigments

For the photosynthetic pigment assay, samples of 2 mL cell suspension were harvested by centrifugation at 6000× *g* for 7 min, supernatant was discarded, and 2 mL of pre-cooled methanol (−4 °C) was added to the pellet with the alginate. The mixture was placed in darkness at 4 °C and incubated for 60 min or longer until the precipitate became white. The absorbance of the supernatant was measured at 470 nm, 665 nm, 720 nm.

Chl a and total carotenoid concentrations were calculated according to the following formulas:Chl a (μg/mL) = 12.9447 × (A665 − A720)
Total carotenoids (μg/mL) = [1000 × (A470 − A720) − 2.86 × (Chl a (μg/mL))]/221

### 2.8. Single-Cell Raman Spectrum Analysis

A portion of the cell mixture was dewatered using a centrifuge (5810R, EPPENDORF, Hamburg, Germany) at 6000 rpm for 5 min. The microalgal pellet was rinsed and resuspended with deionized water for three times. The resuspended algal liquid was sucked into a capillary (50 mm length × 1 mm width × 0.1 mm height, Camlab, Cambridge, UK) and placed on a slide. The slide was placed on the motorized stage of Horiba Raman spectrometer (Horiba LabRAM HR Evolution, HORIBA Scientific, Northampton, UK). A 100× magnifying dry objective (NA = 0.9, Olympus Co., Ltd., Southend-on-Sea, UK) was used for sample observation and Raman spectra acquisition. The Raman scattering from the cells was excited by a 532 nm laser with an approximate laser power of 100 mW, and the collection time was 10 s per spectrum. Each sample was randomly selected from 20 microalgal cells and 4 blank regions to collect Raman spectra.

### 2.9. Raman Spectra Analysis

All Raman spectra were recorded, combined, smoothed and baseline corrected by LabSpec 6. Spectra smoothing was accomplished using the polynomial function (Savitzky–Golay) method. The positions of the spectra bands were determined using a GaussLor constructor and subsequently imported into Origin 2021 for further analysis.

## 3. Results

### 3.1. Screening Results of Coral Symbiotic Microorganisms

We obtained 41 cultivable bacterial isolates associated with the corals *A. hyacinthus* and *G. fascicularis* by 5 different media. These bacteria were identified as belonging to various genera, including *Vibrio* sp., *Ruegeria* sp., *Bacillus* sp., *Thalassotalea* sp., and others such as *Microbulbifer* sp. (Table 1). The quantity and species diversity of the strains varied across the different culture media, with the most diverse strains identified in the MA2216E medium. It was found that *Ruegeria* sp. was the dominant strain, which outnumbered *Vibrio* sp. among all the cultivable bacteria. We further investigated the nitrogen fixation potential of these strains by amplifying specific gene sequences. Among the three *Ruegeria* sp. strains, *Ruegeria* sp. MR31c was positive for nirK and nifH, *Ruegeria conchae* was positive for nirS, and *Ruegeria lacuscaerulensis* was positive for nifH, respectively. Subsequently, we selected the *Ruegeria* sp. MR31c strain for further investigation. The phylogenetic relationship among these three stains of *Ruegeria* sp. was shown in Appendix A. At the same time, we isolated a strain of *Symbiodinium* sp. that was identified as *Durusdinium* sp.

### 3.2. Physiological and Photosynthetic Response to Co-Cultivation of Symbiodiniaceae and Bacteria

The photosynthetic efficiency of *Symbiodiniaceae*, with the treatment of varying *Ruegeria* sp. MR31c concentrations, was evaluated by chlorophyll fluorescence in vivo and by the accumulation of main photosynthetic pigments. The *Durusdinium* sp. growth was observed to be influenced by both nutrient availability and co-cultured *Ruegeria* sp. MR31c concentrations. To account for potential issues arising from excessive bacterial concentration, such as an overabundance of nitrogen that can destabilize the coral–*Durusdinium* sp.–bacterial symbiosis, and medium turbidity that causes shading, the maximum co-culture concentration selected by the laboratory was OD_600_ = 0.5 [34].

Figure 1 illustrates the impact of probiotic bacterium *Ruegeria* sp. MR31c on the *Durusdinium* sp. growth alone and bacteria-added batch cultures vs. time and bacterial concentration (represented by Fv/Fm). As shown in Figure 1A, in the culture without *Ruegeria* sp. MR31c (as the control), it was found that the Fv/Fm of *Durusdinium* sp. decreased within the first 7 days of culture from 0.062 to a minimum of 0.012, then the value of Fv/Fm was up to 0.033 by day 7 to 14. In 14-day mixed cultures with low *Ruegeria* sp. MR31c concentrations (the optical density OD600 = 0.005 and 0.01), Fv/Fm values of *Durusdinium* sp. were similar to those of the control. While it was observed that when the bacterial cell density achieved OD_600_ > 0.01, Fv/Fm of *Durusdinium* sp. increased proportionally with the increasing bacteria concentrations. Particularly, when the optical density of *Ruegeria* sp. MR31c was up to OD_600_ = 0.5, Fv/Fm of *Durusdinium* sp. increased from 0.089 to 0.276 between the 6th and 14th day of culture, indicating that the addition of probiotic *Ruegeria* sp. MR31c at sufficiently high concentrations significantly enhanced the maximum quantum efficiency of PS II in *Symbiodiniaceae* and promoted the cell growth.

The relative accumulations of chlorophyll a (Chl a) and total carotenoids were measured specrophotometrically in 14-day algal cultures under nitrate-depleted conditions. We found that both the Chl a and total carotenoids of *Durusdinium* sp. grown alone were significantly reduced under nitrate-depleted conditions compared to *Ruegeria* sp. MR31c-added conditions (Figure 1B). However, when the optical density of bacteria OD_600_ was less than or equal to 0.01, the relative ratio of Chl a and total carotenoids showed no significant changes, as observed during the 14-day culture. As depicted in Figure 1C, when the optical density (OD_600_) of *Ruegeria* sp. MR31c increased from 0.05 to 0.5, the mixed cultures appeared darker, with a yellowish-brown color. In contrast, when the bacteria OD_600_ was less than 0.01, the color of the co-cultures remained essentially unchanged. Additionally, cultures with bacteria only remained grayish-white in the 14-day culture, irrespective of *Ruegeria* sp. (Figure 1D). This phenomenon suggests that low concentrations of bacteria could not rescue the algae from nutrient deprivation and may even lead to algal starvation and death in nitrogen-free medium, whereas a higher concentration of bacteria may provide organic nutrients to support algal growth. It is important to note that our current experiments indicated that while higher concentrations of the symbiotic microbes can meet the nutritional needs of *Symbiodiniaceae* in oligotrophic environmental conditions, excessive growth of certain symbionts could disrupt the coral–*Symbiodiniaceae*–microbial equilibrium, as symbionts may compete for substrates and nutrients. The existing literature also highlights the importance of controlling symbiont populations within the hosts to maintain the symbiotic relationship and prevent adverse consequences, such as blenching and pathogen infection [35].

### 3.3. Raman Spectra of Single Microalgal Cell

Averaged Raman spectra for the single cell *Durusdinium* sp. is shown in Figure 2, with the corresponding biological band assignments detailed in Table 2. These spectra were generated using a laser power of 100 mW with an exposure time of 1 s. To ensure sufficient sample coverage at each time point, we followed a sampling strategy to collect a total of 60 SCRS per time point, as demonstrated in a prior study by He et al. [36]. Therefore, we selected 20 cells in each of the triplicate cultures to test their SCRS. Figure 2 displays *Durusdinium* sp. cells within the capillary tube, observed under a 100× objective microscope. The acquired spectral range covered from 300 to 3500 cm^−1^. The characteristic peaks determined from the literature [36,37,38] for abundant cellular components such as lipids, carbohydrates, proteins, and nucleic acids were clearly visible in the spectra. Noteworthy peaks include the ring breathing at 655.246 cm^−1^, amide III random at 1269.12 cm^−1^, and the C-N stretching at 1130.31 cm^−1^.

### 3.4. Raman Spectra of Single Bacterial Cell

On calcium fluoride (CaF_2_) microscope slides, a laser at the wavelength of 532 nm was employed to randomly pick and measure 60–80 bacterial cells. The spectral range spanned from 394.11 to 3540.90 cm^−1^. The Raman spectrum of *Ruegeria* sp. MR31c is shown in Figure 3, with peaks assigned to the intracellular components, such as amino acids, carbohydrates, proteins and lipids. The biological assignment of bands corresponding to the Raman spectrum is listed in Table 3.

### 3.5. SCRS Dynamics of Forward ^15^N-Labelling in the Culture of Ruegeria sp. MR31c

Recently, ^15^N-stable isotope probing–single cell Raman spectroscopy (^15^N-SIP-SCRS) has been developed for cellular research. SCRS offers superior signal quality compared to fluorescent signals for revealing bacterial, metabolic and regulatory information, as it provides insight into intracellular structures. Isotope labeling involves replacing one or more atoms in a molecule or compound with isotopes of the same element but with a different atomic mass. When isotopes are introduced into a molecule or compound, it affects the mass of the atoms involved in the vibrations, subsequently alters the vibrational frequencies. This often leads to a red shift in Raman spectra, where the Raman peaks are shifted to lower energy or longer wavelengths compared to the spectra of the non-isotope-labeled molecule. The extent of the shift depends on the specific atoms involved and the positions at which isotopic substitution occurs. It has recently been found that some nitrogen-associated bands of bacterial SCRS shifted to a lower or higher wavelength after *Ruegeria* sp. MR31c cells were fed with isotope-labeled ^15^N-nitrogen substrates [39]. To investigate the impact of ^15^N-incorporation on bacterial metabolism using Raman spectroscopy, we cultured *Ruegeria* sp. MR31c on L1 medium with either ^14^N- or ^15^N-NH_4_Cl as the sole source of nitrogen. Centrifuged and rinsed ^15^N-labeled bacteria were inoculated into sterile, nitrogen-free *Durusdinium* sp. growth at a density of 109 cells mL^−1^ as the mixed culture for sampling. A comparison of these spectra revealed red shifts in the characteristic regions, with all shifted bands originating from nitrogen-associated compounds, as shown in Figure 4. In 48 h of incubation, the original ^14^N in the *Ruegeria* sp. MR31c cells was gradually replaced by the fed ^15^N, causing forward shifts from ^14^N Raman bands at 824, 958, 1375, and 1460 cm^−1^ back to ^15^N Raman bands at 804, 920, 1364 and 1448 cm^−1^. This red-shift phenomenon in the Raman spectra indicates that ^15^N-labelling substrate has participated in the biosynthesis of intracellular cytoskeletons for *Ruegeria* sp. MR31c. Among them, bands at 824 and 958 cm^−1^ were associated with the amino acids, tyrosine and proline, respectively, which have been well demonstrated by Thmos et al., Mary et al., and also supported by other bacterial SERS works [40,41]. Tyrosine and proline both contain amino groups situated on branched chains and ring structures, which accounts for the significant red shift in 824 and 958 cm^−1^ band when ^15^N substitutes ^14^N in these amino acids. Meanwhile, the band from 1375 cm^−1^ was assigned to nucleotide bases (thymine, adenine, guanine), and displayed a shift of 11 cm^−1^. Guanine and adenine contain four nitrogen atoms in their ring structure and one nucleobase in the side chain, respectively, and thymine contains two nitrogen atoms in its ring structure, contributing to the observed approximately 11 cm^−1^ corresponding shifts. Furthermore, the protein maker band shifted 12 cm^−1^ from ^14^N-SCRS at 1460 cm^−1^ to ^15^N-SCRS at 1448 cm^−1^. This displacement suggests that ^15^N-ammonium is infiltrated into the cell to participate in protein synthesis.

### 3.6. Dynamics of SCRS Characteristic Bands in Algal and Bacteria Co-Culture

Figure 5 shows all the shifted SCRS bands induced by ^15^N assimilation in the *Symbiodiniaceae*–bacterial co-culture. Among them, the 655 cm^−1^ band was assigned to guanine-related biomolecules, displaying a small shift of 7 cm^−1^. Guanine, containing four nitrogen atoms in its ring structure, contributed to the observed 15N-corresponding shift in the Raman spectra. The band at 1130 cm^−1^ (Cytochrome c) and 1270 cm^−1^ (lipids) were assigned to the C-N stretch and amide III random, exhibiting shifts of 7 cm^−1^ and 14 cm^−1^, respectively, and providing a potential goal to incorporate ^15^N-NH_4_Cl stable isotope into C-N. Cytochrome c (Cyt c) is a cytosolic encoded hemoglobin involved in photosynthetic electron transport. The heme b (Fe-protoporphyrin IX) is always attached to the Cyt c via a CX2CH motif, and the polar amino acid histidine is coordinated to the heme iron ion and two cysteines, with their sulfhydryl groups bound to heme by two thioether bonds. Both histidine and cysteine are polar amino acids distributed on the outer surface of the Cyt c, which is approximately spherical in shape [42]. Histidine contains three nitrogen atoms in its ring structure and one nitrogen atom in the side alkyl chain, and cysteine contains one nitrogen atom in the side alkyl chain. These characteristics explain the significant shift in the 1130 cm^−1^ band when ^15^N substitutes ^14^N in Cyt c in Figure 5. The shift of the SCRS from 1270 cm^−1^ to 1256 cm^−1^ in Raman spectra suggests that bacteria may be involved in the synthesis of unsaturated fatty acids during the growth of Durusdinium sp. A majority of cellular nitrogen was also found in protein-containing amide groups. The SCRS band from amide III of protein-related (1240 cm^−1^) displayed a large shift of 12 cm^−1^. The 942 cm^−1^ band was assigned to C-O stretching, C-O-C and the C-O-H deformation of starch. The shift from 942 cm^−1^ to 927 cm^−1^ suggests that ^15^N may be involved in material and energy transformations during photosynthetic respiration when it enters *Symbiodiniaceae* cells.

Surprisingly, protein-associated shifts were not observed in SCRS bands of 1008 and 1643 cm^−1^. The 1008 cm^−1^ band was assigned to the benzene ring breathing vibration of phenylalanine. Since the only nitrogen in phenylalanine is located on the side alkyl chain of the benzene ring, its effect on ring breathing vibration is likely to be minimal, explaining the lack of shifts. The SCRS band of 1643 cm^−1^ was assigned to the amide I vibration associated with the secondary protein structure of α-helix and β-sheet, and thus could be not sensitive enough to ^15^N substitution. Notably, 972 (C-C wagging), 1090 (C-O stretching) and 1446 cm^−1^ (CH2, CH3 bending), that were not involved in nitrogen metabolism, displayed no shifts with ^15^N substitution in biomass. Despite the limited sensitivity of the nitrogen-related bands following ^15^N assimilation, significant alterations in nucleotide bases and other molecules still hold the potential to enhance our comprehension of crucial biochemical processes. It is noteworthy that, from the above spectral analysis, the combination of isotope-labeling and single Raman spectroscopy, or ^15^N-stable isotope probing–single cell Raman spectroscopy (^15^N-SIP-SCRS), has demonstrated to be a powerful technique to visualize the isotope transformation process from the labeled substrate into intracellular substances of *Ruegeria* sp. MR31c, and subsequently into synthesized cytoskeletons of *Durusdinium* sp.

## 4. Discussion

The emerging evidence suggests that specific bacterial taxa and *Symbiodiniaceae* interactions may be crucial to uphold holobiont metabolic functioning [43,44]. However, the precise impact of bacteria on *Symbiodiniaceae* and their potential contribution of nitrogen-related substances to *Symbiodiniaceae* have remained elusive. Here, we found that the co-culture of *Durusdinium* sp. with native, potentially beneficial bacteria induced significant alterations in the photosynthetic system and pigment accumulation, which coincided with changes in cell color. Consequently, we conducted further investigations to determine whether the bacteria supply N-containing compounds to support *Durusdinium* sp. growth based on ^15^N-SIP-SCRS.

In our research, it was found that the most abundant and culturable core members of coral-associated bacterial communities were *Ruegeria* sp., followed by *vibrio* sp.*,* and the genus of α-proteobacteria and γ-proteobacteria, respectively. *Ruegeria* sp. has surged in scientific attention lately, as its ubiquity under eutrophic conditions and its denitrification potential make this type of marine bacteria promising candidates for the use of probiotic applications to enhance coral health. However, our understanding of the functional role of *Ruegeria* sp. in the coral holobiont and its interaction with other holobiont members remains limited. Notably, the *Vibrio* species have been implicated in causing diseases in marine organisms, with *Vibrio coralliilyticus* identified as a pathogen responsible for coral bleaching and tissue lysis [45]. In recent studies, the relative abundance of *Vibrio* increased in coral colonies at higher temperatures, while the relative abundance of *Ruegeria* sp. decreased, indicating that the occurrence of coral disease might be linked to the decrease in *Ruegeria* sp. in abundance during sea surface temperature elevation [46]. Meanwhile, *Ruegeria* sp. was seen to play vital roles in marine ecosystems by supplying vitamin B_12_ to plankton and contributing to carbon and sulfur cycles [18]. Hence, *Ruegeria* sp., which belong to the Rosebacter clade, are not only potential probiotics for *Symbiodiniaceae*, but also integral members of coral symbiotic systems.

One of the predominant characteristics exhibited by many unicellular algae subjected to N-deprival is a reduction in photosynthetic activity [47]. This phenomenon is substantiated by the findings presented in Figure 6. During N limitation, decreased chlorophyll content and photosynthetic function in microalgae, which may be related to the decreased levels of transcripts and proteins associated with photosynthetic activity [48]. Additionally, N deficiency also typically results in ease of reproduction and accumulated photosynthate [49]. By contrast, co-culturing *Symbiodiniaceae* cells with bacterial solution exhibited minimal differences in PSII activity, Chl a level, and total carotenoids compared to nutrient-replete conditions, likely due to the bacterial conversion of atmospheric nitrogen into intracellular compounds, such as ammonium, for other organisms’ utilization (Figure 6) [43,44]. In fact, we found that the photosynthetic capacity and pigment accumulation during the mixed *Symbiodiniaceae*–bacterial culture were lower than in the N-sufficient L1 medium, implying some degree of limitation on algal growth. This limitation might be attributed to the turbidity caused by bacterial growth and the relatively high quantity of bacteria attached to the surface of the algal cells, resulting in a reduced light intensity of less than 60 μmol of photons m^−2^ s^−1^. On the other hand, changes in the bacteria metabolism presumably led to a reduced availability of N for the algae, resulting in a slower rate of algal cell growth.

Our study demonstrated the efficacy of combining Raman spectroscopy with ^15^N-stable isotope labeling (^15^N-SIP-SCRS) in tracing the labeled nitrogen source to be transported from one species to another. In more detail, ^15^N was infiltrated into bacterial cells in the culture medium to participate in the intracellular metabolic activities. The nitrogen source was then found in the macromolecules, such as carbohydrates, proteins and lipids, inside *Symbiodiniaceae* cells. ^15^N assimilation in cells induced a clear red shift in SCRS. Cui et al. probed the nitrogen assimilation by bacteria at single-cell level, which showed multiple distinguishable SCRS band shifts and displayed a linear relationship with ^15^N content. This shift was evident in multiple distinguishable SCRS band shifts, including bands at 824 and 958 cm^−1^, which were assigned to tyrosine (plane ring breathing) and proline (C-N stretching), suggesting that ^15^N was incorporated into bacterial intracellular amino acid synthesis. Tyrosine is a commercially important compound, as it is widely used in food, chemical, pharmaceutical and cosmetic industries [50]. In bacteria, tyrosine is usually the end product of its biosynthetic pathway [51]. Proline is a natural penetrant and antioxidant, affecting numerous metabolic processes in organisms, such as signaling, stress protection and energy production [52]. Proline metabolism associated with the TCA cycle impact NADP^+^/NADPH levels, thereby driving the pentose phosphate pathway and promoting nucleotide synthesis and cell division [53]. Amino acids like tyrosine and proline serve as the fundamental building blocks of biological macromolecules, which makes it easy to explain that the band at 1240 cm^−1^ attributed to the protein has shifted by 12 cm^−1^. Moreover, the band at 1375 cm^−1^ was assigned to nucleotide bases (thymine, adenine, guanine), exhibiting a shift of 11 cm^−1^, which can be attributed to the unique molecular structure of the bases. It has been reported that adenine, in particular, has a greater potential to induce larger shifts in the Raman spectrum compared to guanine and cytosine, which could be involved in adenine moiety for the production of co-enzymes or co-factors (e.g., ATP, NAD, NADP and FAD) in cells. They are thus involved in a variety of anabolic reactions (e.g., tricarboxylic acid cycle and Calvin cycle). Therefore, ^15^N may have more opportunities to be doped into adenine-containing molecules [42].

Our research provides insights into the crucial role of bacterial contributions in supplying nitrogenous compounds to *Durusdinium* sp. in the *Symbiodiniaceae* loop, which are integral for the synthesis of essential nutrients within the cellular structures of microalgae. Nitrogen metabolism in microalgae is of critical importance for their growth [54]. In environments where an adequate inorganic nitrogen source is available, inorganic salts traverse the cytosol via transporter proteins and undergo reduction to ammonium salts through the action of reductase enzymes. Subsequently, glutamine is synthesized, participating in amino acid metabolism. In addition to inorganic salts, microalgae can also utilize organic nitrogen sources such as urea, purines, and some amino acids [55,56]. In our study, the band from 655 cm^−1^ to 649 cm^−1^ was assigned to the ring breathing of guanine-related biomolecules, indicating that the purines produced by bacterial metabolism were utilized by *Durusdinium* sp. Meanwhile, the bands at 942 cm^−1^ and 1270 cm^−1^ were assigned to C-O/C-O-C/C-O-H stretching and Amide III random vibrations, respectively, related to starch and lipids. Microalgae demonstrate exceptional efficiency in converting solar energy and carbon dioxide into intracellular energy-dense macromolecules, which mainly includes carbohydrates, proteins and lipids [57,58]. Starch and lipid synthesis in microalgae occurs concomitantly, utilizing carbon precursors produced during the Calvin cycle. This intricate process necessitates the involvement of coenzymes and cofactors [59]. The shifts observed in the peaks associated with starch and lipids may be attributed to the synthesis of co-enzymes implicated in the TCA cycle following the assimilation of purines into the cells. In addition, the intensity of the C-N stretching band at 1130 cm^−1^ decreased, and the corresponding ^15^N-substituted band emerges at 1123 cm^−1^. Similarly, the protein band at 1240 cm^−1^ disappeared, while a new band appeared at 1228 cm^−1^. Cyt c is also a cytosolic encoded hemoglobin involved in photosynthetic electron transport. The protein is bound to the heme group via amino acid, and the distortion of the heme group by the surrounding protein leads to characteristic spectra. We hypothesize that the red shift of the two bands on proteins is due to the protein synthesis using the amino acids transported from the symbiotic bacteria. Most dinoflagellates are difficult to grow in pure cultures without associated bacteria, exemplified by *Pfisteria*, which was unable to grow or even perished in a sterile medium, while the addition of the symbiotic bacterium α-proteobacteria strain restored its normal growth [60]. This phenomenon suggests that the bacteria present in dinoflagellate cultures provide the necessary components for the successful growth of dinoflagellates outside the coral host. Furthermore, corals acquire bacteria during early developmental stages, preferentially uptake members of the Roseobacter clade, demonstrating that bacteria may play a role in the early stages of coral development and coral colonization [61]. The characterization of Symbiodinium species from different oceans showed that Roseobacter branch is closely related to the Symbiodinium clade, being detected in clade A–F [62]. Additionally, *Symbiodiniaceae* displayed selective associations with specific microbiota. In cases such as Chlorella and Scenedesmus, the bacteria with the most active transcription of key genes associated with plant–microbe interactions belong to the phylum of α-proteobacteria [63].

## 5. Conclusions

In this study, we investigated interactions between *Symbiodiniaceae* and beneficial bacteria, with the hypothesis that there exists an inner symbiotic loop of *Symbiodiniaceae* and bacteria within the coral symbiotic loop, and the metabolic exchanges among them facilitate the essential nutritional interplay required for the *Symbiodiniaceae* cellular growth and proliferation.

The nitrogen-fixing bacterium strain *Ruegeria* sp. and the *Symbiodiniaceae* strain *Durusdinium* sp. were selected to study how the beneficial bacterium enables *Symbiodiniaceae* to thrive in a nitrogen-deficient environment. Our findings validated the hypothesis and substantiated the significant role of potentially beneficial bacteria in fostering the growth of *Symbiodiniaceae* within isolated cultures. Moreover, it underscores the capacity of bacteria to supply inorganic nitrogen sources in vitro, thereby supplying nitrogen-containing compounds to *Symbiodiniaceae* and actively participating in the nitrogen cycle.

Our research results indicated that ^15^N-SIP-SCRS is a novel method for revealing the dynamic features of the inter-cellular transport of nutrients, based on the combination of the labeling of the nitrogen source by ^15^N-SIP and the identification of intracellular biomolecules present in individual cells by Raman spectroscopy. The technique enabled the uptake of the labeled nitrogen source by the beneficial bacterium to be traced, and eventually the labeled substrates were integrated into the intracellular components of *Symbiodiniaceae* cells. It thus supported our assumption that metabolites function as a nutritional currency for the symbiotic systems.

We believe that beneficial reciprocity needs to be shown within the coral holobiont, and further research endeavors should aim to employ labeling techniques for symbiotic bacteria, *Symbiodiniaceae* and coral, to demonstrate the multi-directional benefits of their symbiosis. In the interim, we are quantifying key cellular metabolites to facilitate non-destructive, unlabeled metabolic flow studies. Future investigations into coral–*Symbiodiniaceae*–bacteria interactions are needed for a comprehensive understanding of the intricate coral reefs.

## Figures and Tables

**Figure 1 microorganisms-12-01217-f001:**
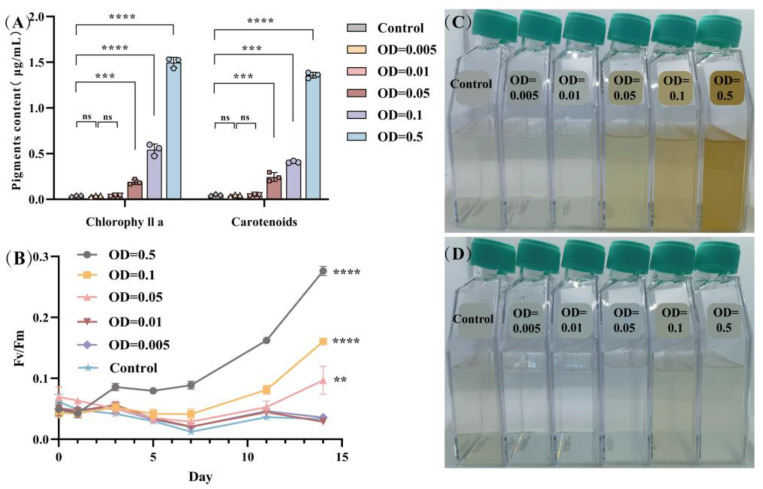
Photosynthetic function of *Durusdinium* sp. cells growing in flask. (**A**) chlorophyll a levels and total carotenoids for *Durusdinium* sp. cell density vs. *Ruegeria* sp. MR31c concentrations. (**B**) Maximum PS II efficiency Fv/Fm, for *Durusdinium* sp. Cell density vs. *Ruegeria* sp. MR31c concentrations. (**C**) Mixed culture solution with *Ruegeria* sp. MR31c concentrations. (**D**) Probiotic *Ruegeria* sp. MR31c cell culture. Means ± SDs for three independent trials are shown with the *p*-values (*t*-test) for the probabilities that the differences are significant. (ns, *p* > 0.05; ** *p* < 0.01; *** *p* < 0.001; **** *p* < 0.0001).

**Figure 2 microorganisms-12-01217-f002:**
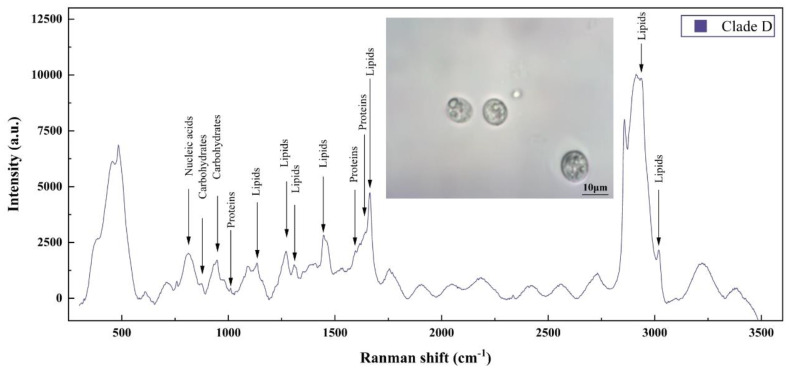
Raman spectra of *Durusdinium* sp. with 100× objective imaging.

**Figure 3 microorganisms-12-01217-f003:**
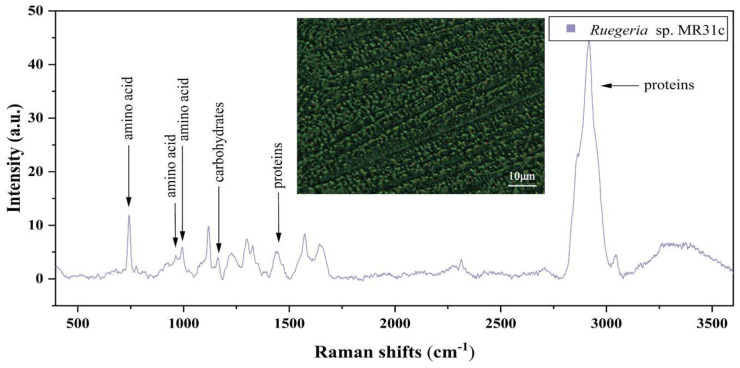
Raman spectrum of individual of *Ruegeria* sp. MR31c cell, with 100× objective imaging.

**Figure 4 microorganisms-12-01217-f004:**
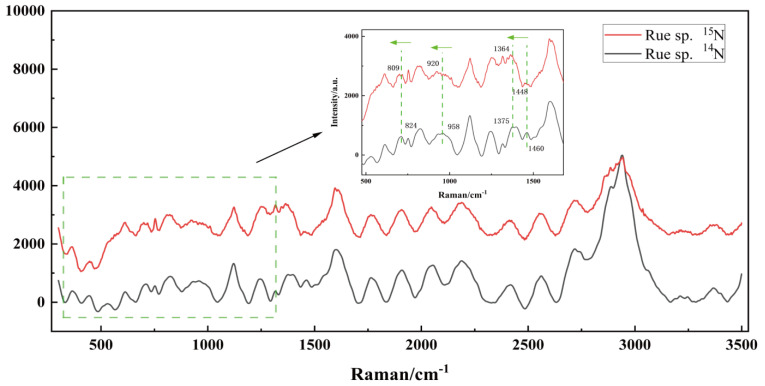
“red-shift” of SCRS by the comparison of ^15^N- (red or light line) and ^14^N- (black or dark line) SCRS with the bacterium *Ruegeria* sp. MR31c grown in L1, in which the ^15^N labeled NH_4_Cl was the sole nitrogen source. The arrow indicates the dynamic change of the "red shift" of the *Ruegeria* sp. resonance Raman signal.

**Figure 5 microorganisms-12-01217-f005:**
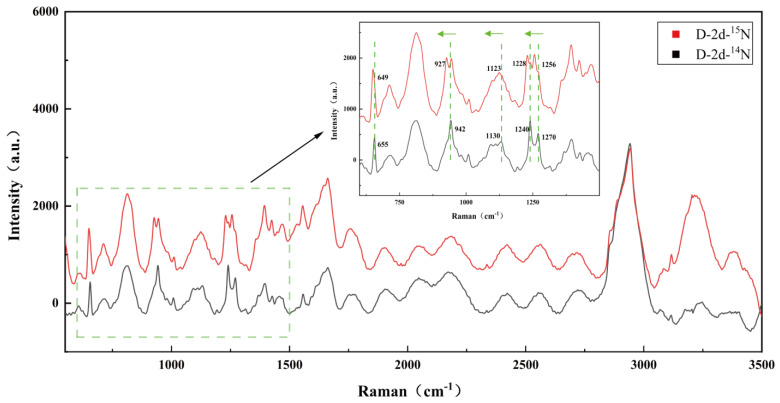
“red-shift” of SCRS by the comparison of ^15^N- (red or light line) and ^14^N- (black or dark line) SCRS with *Durusdinium* sp. single cell from the mixture culture of *Symbiodiniaceae* with *Ruegeria* sp. MR31c in fresh L1 medium. The arrow indicates the dynamic change of the "red shift" of the *Durusdinium* sp. resonance Raman signal.

**Figure 6 microorganisms-12-01217-f006:**
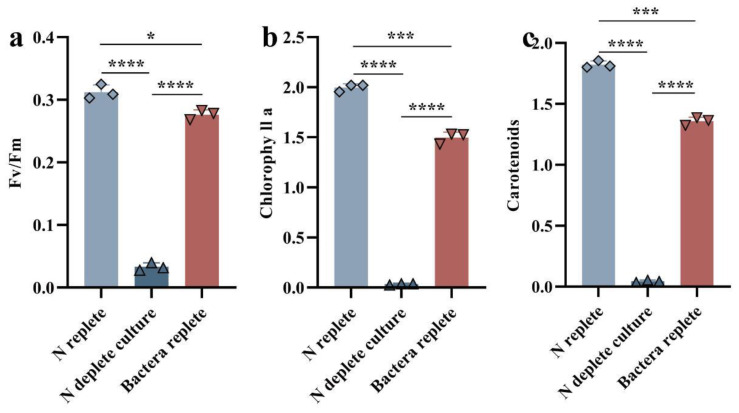
Photosynthetic function in *Durusdinium* sp. cells growing in different conditions. (**a**–**c**) Fv/Fm, chlorophyll a levels and total carotenoids were determined for *Durusdinium* sp. cells at steady state in N-deplete culture, N-replete culture, as well as for bacterial N-replete cultures, as described in Materials and Methods. Means ± SDs for three independent trials are shown with the *p*-values (*t*-test) for the probabilities that the differences observed are significant. (* *p* < 0.05; *** *p* < 0.001; **** *p* < 0.0001).

**Table 1 microorganisms-12-01217-t001:** 41 culturable symbiotic microbial strains isolated from coral reefs.

Genus	No.	Name	Genbank Accession Number	Medium	Affiliated Coral Types	Identity
*Vibrio*	JW-1	*Vibrio owensii*	CP045859.1	LB; L1	*A. hyacinthus*	(1412/1412) 100%
	JW-2	*Vibrio vulnificus*	MN860081.1	LB	*A. hyacinthus*	(1448/1448) 100%
	JW-3	*Vibrio coralliilyticus*	CP031472.1	LB	*A. hyacinthus*	(1394/1408) 99%
	JW-4	*Vibrio* sp. *Strain* JC009	CP092106.1	L1	*A. hyacinthus*	(1452/1452) 100%
	JW-5	*Vibrio alginolyticus*	CP054700.1	MA2216E	*G. fascicularis*	(1389/1390) 99.9%
	JW-6	*Vibrio rotiferianus*	AP019798.1	MA2216E	*G. fascicularis*	(1415/1416) 99.9%
*Ruegeria*	JW-7	*Ruegeria conchae*	CP031472.1	MA2216E	*A. hyacinthus*	(1358/1358) 100%
	JW-8	*Ruegeria* sp. MR31c	HQ439523.1	MA2216E	*A. hyacinthus*	(1334/1348) 99%
	JW-9	*Ruegeria* sp. atlantica	MW828512.1	MA2216E	*A. hyacinthus*	(1308/1308) 100
	JW-10	*Ruegeria* sp. LR4	KU560503.1	MA2216E	*A. hyacinthus*	(1341/1354) 99%
	JW-11	*Ruegeria* sp. strain MP15.1	OQ435566.1	MA2216E	*A. hyacinthus*; *G. fascicularis*	(1317/1330) 99%
	JW-12	*Ruegeria arenilitoris*	MG896151.1	MA2216E	*G. fascicularis*	(1309/1322) 99%
	JW-13	*Ruegeria lacuscaerulensis*	MH283799.1	L1	*G. fascicularis*	(1432/1446) 99%
*Bacillus*	JW-14	*Bacillus horikoshii*	DQ289065.1	MA2216E	*G. fascicularis*	(1434/1448) 99%
	JW-15	*Bacillus weihaiensis*	CP016020.1	MA2216E	*G. fascicularis*	(1434/1434) 100%
	JW-16	*Bacillus coahuilensis*	EF014447.1	MA2216E	*G. fascicularis*	(1449/1449) 100%
*Thalassotalea*	JW-17	*Thalassotalea euphylliae*	MW828496.1	L1	*A. hyacinthus*; *G. fascicularis*	(1379/1392) 99%
	JW-18	*Thalassomonas loyana*	HQ439553.1	MA2216E	*G. fascicularis*	(1408/1422) 99%
	JW-19	*Thalassomonas agarivorans*	HQ439504.1	MA2216E	*G. fascicularis*	(14111425) 99%
*Thalassospira*	JW-20	*Thalassospira* sp. 2ta1	FJ952805.1	MA2216E	*A. hyacinthus*	(1361/1374) 99%
*Microbulbifer*	JW-21	*Microbulbifer* sp. Alg-AMLN-14-8	MK453424.1	MA2216E	*A. hyacinthus*	(1406/1406) 100%
*Phaeobacter*	JW-22	*Phaeobacter* sp. strain 088	MK801649.1	MA2216E	*A. hyacinthus*	(1333/1333) 100%
*Alteromonas*	JW-23	*Alteromonas aestuariivivens*	NR157790.1	L1	*A. hyacinthus*	(1457/1517) 96%
	JW-24	*Alteromonas macleodii*	OX359243.1	L1; CDA	*A. hyacinthus*; *G. fascicularis*	(1394/1408) 99%
*Roseovarius*	JW-25	*Roseovarius* sp.	MZ262971.1	MA2216E	*A. hyacinthus*	(1311/1324) 99%
*Roseobacter-* *aceae*	JW-26	*Shima* sp. LR11	KU560500.1	NSW	*A. hyacinthus*	(1352/1352) 100%
	JW-27	*Shimia isoporae*	MH283808.1	L1	*G. fascicularis*	(1355/1355) 100%
*Marinobacter*	JW-28	*Marinobacter* sp.	MT210870.1	MA2216E	*A. hyacinthus*	(1503/1503) 100%
*Labrenzia*	JW-29	*Labrenzia* sp.	MK493531.1	MA2216E	*A. hyacinthus*	(1389/1403) 99%
*Psychrosphaera*	JW-30	*Psychrosphaera* sp.	MZ262895.1	L1	*A. hyacinthus*	(1385/1385) 100%
*Microbacterium*	JW-31	*Microbacterium esteraromaticum*	MT453933.1	MA2216E	*G. fascicularis*	(1393/1393) 100%
	JW-32	*Microbacterium* sp. OB57	JN942151.1	MA2216E	*G. fascicularis*	(1418/1432) 99%
*Rossellomorea*	JW-33	*Rossellomorea aquimaris*	MK256784.1	MA2216E	*G. fascicularis*	(1451/1451) 100%
*Tropicibacter*	JW-34	*Tropicibacter* sp.	MK801651.1	MA2216E	*G. fascicularis*	(1336/1336) 100%
*Stutzerimonas*	JW-35	*Stutzerimonas stutzeri*	MT356167.1	CDA	*G. fascicularis*	(1461/1475) 99%
*Acinetobacter*	JW-36	*Acinetobacter seifertii*	OP114754.1	CDA	*G. fascicularis*	(1409/1423) 99%
	JW-37	*Acinetobacter soli*	OP854766.1	CDA	*G. fascicularis*	1403/1403 100%
*Enterobacter*	JW-38	*Enterobacter cancerogenus*	CP025225.1	CDA	*G. fascicularis*	(1406/1406) 100%
*Marinomonas*	JW-39	*Marinomonas* sp.	MG099520.1	CDA	*G. fascicularis*	(1462/1476) 99%
*Aerococcus*	JW-40	*Aerococcus viridans*	MT502756.1	MA2216E	*G. fascicularis*	(1423/1437) 99%
*Pseudoalteromonas*	JW-41	*Pseudoalteromonas shioyasakiensis*	KU321310.1	MA2216E	*G. fascicularis*	(1407/1421) 99%

**Table 2 microorganisms-12-01217-t002:** The reference Raman bands highly correlated with *Symbiodiniaceae.*

Component	Raman Bands (cm^−1^)	Assignment
unknown	655.246	v (C-S) gauche
unknown	754.235	Symmetric breathing of tryptophan
nucleic acids	810.75	C-O-P-O-C in RNA backbone
carbohydrates	872.62	C-C stretching, Hydroxyproline
carbohydrates	943.17	C-O stretching; C-O-C and C-O-H deformation; α-helix C-C backbone
lipids	972.30	V (C-C) wagging
proteins	1008.68	C-C aromatic
unknown	1090.82	C-O stretching
proteins	1130.31	C-N stretching
lipids	1269.12	Amide III random, lipids
lipids	1305.86	CH_3_/CH_2_ twisting or bending mode of lipids
unknown	1364.78	*vs* (CH_3_) Adenine, guanine, tyrosine, tryptophan
unknown	1405.91	v (COO-)
lipids	1446.82	CH_2_, CH_3_ bending modes
proteins	1595.28	C=N and C=C stretching in quinoid ring
proteins	1609.89	Cytosine (NH_2_)
proteins	1643.04	Amide I band (protein band)
lipids	1663.69	(C=C) cis, lipids, fatty acids
lipids	2857.48	CH_2_ symmetric stretch of lipids
lipids	3017.76	v=CH of lipids

**Table 3 microorganisms-12-01217-t003:** The reference Raman bands highly correlated with *Ruegeria* sp. MR31c.

Component	Raman Bands (cm^−1^)	Assignment
amino acid	743.26	C-S stretch
amino acid	958.87	C-N stretching
amino acid	824.35	aromatic ring vibration
amino acid	994.42	C-C aromatic and symmetric ring breath
carbohydrates	1157.26	C-C, C=C band stretch
unknown	1231.47	Amide III, C-N stretch, N-H coupling
unknown	1329.66	DNA, Phospholipids, purine
nucleobase	1375.36	Thymine, adenine, guanine
proteins	1460.45	CH2 bending mode, C-H vibrations
unknown	1573.91	Amide II, nucleic acid, Peptidoglycan
lipids	2918.07	C-H vibrations

## Data Availability

The sequencing data of this study have been uploaded on 15 September 2023 to the Genome Sequence Archive in BIG Data Center (https://ngdc.cncb.ac.cn/?lang=en), Beijing Institute of Genomics (BIG), Chinese Academy of Sciences, with the accession number: PRJCA018758.

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
