# Peer review of "Symbiodiniaceae and Ruegeria sp. Co-Cultivation to Enhance Nutrient Exchanges in Coral Holobiont"

_microorganisms, 2024, doi:10.3390/microorganisms12061217_

Round 1

Reviewer 1 Report

Comments and Suggestions for Authors

In the present study, the authors proposed a hypothesis that there exists an inner symbiotic loop of Symbiodiniaceae and bacteria within the coral symbiotic loop. To prove this hypothesis, different experiments were conducted to demonstrate how metabolic exchanges between Symbiodiniaceae and bacteria facilitate the nutritional supply necessary for cellular growth. As a result, it was observed that the beneficial bacterium, Ruegeria sp. supplied a nitrogen source to the Symbiodiniaceae strain Durusdinium sp., allowing this dinoflagellate to thrive in a nitrogen-free medium. The Ruegeria sp.-Durusdinium sp. interaction was confirmed through 15N-stable isotope probing-single cell Raman spectroscopy, in which 15N infiltrated into bacterial cells for intracellular metabolism. Eventually, the labeled nitrogen source was traced within the macromolecules of Symbiodiniaceae cells. The investigation into Symbiodiniaceae loop interactions validates our hypothesis and contributes to a comprehensive understanding of the intricate coral holobiont. In my opinion, this work presents interesting results and can be published in Microorganisms after minor revision, as follows:

1. Chemical formulas must be adequately revised – some mistakes were found in the text (for example CaCl instead of CaCl2)

2. Considering that the experiments were conducted using 15N, why did the authors not choose mass spectrometry analysis instead of Raman spectroscopy?

3. Item b/Figure 1 – were the experiments conducted in triplicate? Why were not presented error bars? 

4. In my opinion, Raman spectral analysis was not indicative of different components as indicated in Tables 2 and 3 - this profile can be observed in different natural products. MS data must be useful to confirm this hypothesis.  

5. Assignments shown in Tables 2 and 3 must be carefully revised. Some characteristic bands/signals must be included in the attribution since the presented data are so general. Example: amino acid at 958 cm-1 - C-N stretching.

6. What does "plane ring breathing"? Is it correct?

Comments on the Quality of English Language

Minor editing of English language required 

Author Response

Dear Reviewer 1:

Please see the attached word file to view our responses to your constructive comments

Reviewer 2 Report

Comments and Suggestions for Authors

Review of Symbiodiniaceae and Ruegeria sp. Co-cultivation to Enhance Nutrient Exchanges in Coral Holobiont

This paper explores the relationships between Symbiodiniaceae and bacteria in two coral species. The paper is well written and seeks to explain the process without being too lengthy.  This work is critical to the understandings of coral reefs. My comments are minor:

1.       Line 40 check document to make sure the Symbiodiniaceae is italicized where appropriate.

2.       Line 55 change to aiding

3.       Line 95 add ‘been’

4.       Line 99 spell out the full coral names and the reference to the first author to describe this. For example Acropora hyacinthus (Dana 1846)

5.       Line 179 explain more what is mean in 14 day

6.       Line 436 check document and lines for Ruegeria being italicized.

7.       Line 451 change converting. Make sentence shorter.

8.       Line 517 fix the flow of this sentence.

Comments on the Quality of English Language

only needs a few changes.

Author Response

Dear Reviewer 2:

Please see the attached word file to view our responses to your constructive comments
